# On Building and Evaluating a Medical Records Exploration Interface Using Text Mining Techniques

**DOI:** 10.3390/e23101275

**Published:** 2021-09-29

**Authors:** Úrsula Torres Parejo, Jesús Roque Campaña, María Amparo Vila, Miguel Delgado

**Affiliations:** 1Department of Statistics and Operational Research, University of Granada, 18071 Granada, Spain; 2Department of Computer Science and Artificial Intelligence, University of Granada, 18014 Granada, Spain

**Keywords:** knowledge representation, electronic health records, health information systems, content identification, visual interface

## Abstract

Medical records contain many terms that are difficult to process. Our aim in this study is to allow visual exploration of the information in medical databases where texts present a large number of syntactic variations and abbreviations by using an interface that facilitates content identification, navigation, and information retrieval. We propose the use of multi-term tag clouds as content representation tools and as assistants for browsing and querying tasks. The tag cloud generation is achieved by using a novelty mathematical method that allows related terms to remain grouped together within the tags. To evaluate this proposal, we have carried out a survey over a spanish database with 24,481 records. For this purpose, 23 expert users in the medical field were tasked to test the interface and answer some questions in order to evaluate the generated tag clouds properties. In addition, we obtained a precision of 0.990, a recall of 0.870, and a *F*1-score of 0.904 in the evaluation of the tag cloud as an information retrieval tool. The main contribution of this approach is that we automatically generate a visual interface over the text capable of capturing the semantics of the information and facilitating access to medical records, obtaining a high degree of satisfaction in the evaluation survey.

## 1. Introduction

In the medical field, multiple data are collected every day. In order to be useful, data must be processed, which is a complex task [1]. In emergency management, surgical interventions, human resources, and all other hospital areas, it is necessary to extract knowledge from data that would help the general management of the hospital, patient care, and decision making.

As far as textual information is concerned, this is not an easy task since a text may present a large number of syntactic variations or even mistakes. In addition, this information is usually inserted by different people who use different writing patterns. It should also be noted that the number of semantically relevant entities grows constantly and rapidly as new scientific discoveries are made [2,3]. In order to deal with all of these inconveniences and to process information correctly, it is necessary to have intelligent systems that process data semantically, as well as syntactically, which adds structure and facilitate the formulation of semantic queries about textual attributes.

Furthermore, the search for clinical information is becoming a critical technique for rapid and effective access to patients’ information [4], so it is essential to have a simple interface that helps in the query formulation and does not require prior knowledge for its use.

To address these problems, we propose a method to process the textual information contained in these databases by preserving the semantics and offering a multiterm tag cloud as a graphical interface that represents the content of the information enabling the identification of entities and their relationships by allowing related terms to remain grouped together into tags. This graphical interface also permits the user to browse the contents of the database and query it in an intuitive manner.

Tag clouds seem to be a good alternative due to the familiarity that most users have with them as internet navigation tools and their ability to represent information content [5]. They are already used in the medical field for biomedical text summarization, as described in [6], and for content representation, as observed in [7]. However, these authors use a very different generation method to ours, resulting in a mono-term tag cloud, while the tag cloud that we propose is composed of multi-term tags.

The mono-term tag clouds, mostly used in other approaches, produce some important drawbacks regarding the ability to identify contents [8,9,10]. Think, for example, about terms commonly used in medicine such as “central line” or “posterior chamber”; the meaning of these terms is not inferred by the union of the meanings of the words that compose them. Thus, for a correct identification of the information content, these terms must remain grouped within the tag cloud. Another drawback of the mono-term tag clouds is that they do not allow the identification of the relationships between terms, which also affects semantics. Furthermore, we must take into account that whilst the associations between entities usually involve only two entities, in the medical field, relationships may involve more than two (for example, “bone lesion biopsy”), which involves complex associations [11].

On the other hand, one of the main disadvantages attributed to most of tag cloud visualizations proposed by other authors is that they do not have a standardized generation method and that they lack an underlying mathematical model, which is especially useful since it allows the definition of the operations performed in the database [10,12].

To overcome all of these challenges, we propose an automatically generated tag cloud. This tag cloud would allow the user to browse and query the database and facilitate content identification and the relationships between the concepts due to the use of multi-term tags. It is defined mathematically and is obtained by using a standardized method.

### 1.1. Background

#### 1.1.1. Information Extraction in the Clinical Domain

Natural language text processing techniques for information extraction were introduced into the medical field more than two decades ago [13], and they were developed and used for many systems in different applications [14,15].

Information extraction is the task of obtaining structured semantic relationships from unstructured text [16]. The extraction of biomedical information is usually performed with unstructured scientific texts or electronic health records [17]. The main sub-tasks that are carried out include the following: entity recognition, relationship extraction, and event identification [18].

In [19] one of the simplest methods for identifying relationships between entities is applied. They used statistics of co-occurrences to calculate the degree of association between diseases and drugs in clinical records. However, most of the approaches based on co-occurrences achieved high recall and low accuracy: recall is the ability of a model is its ability to find all the relevant cases within a dataset, and accuracy is the fraction between the number of relevant cases found divided by the total number of cases.

The high degree of malformation in medical texts makes searches difficult; therefore, it is convenient to integrate an information retrieval system into the relational databases [20].

In [21,22], we find a new method to process texts and represent them by using intermediate forms that allow the related terms to remain together, thus preserving the semantics and allowing the identification of relationships between the terms. This form of textual representation is also based on co-occurrences, but it is able to obtain high precision values, defining the precision as the fraction of relevant instances among the retrieved instances.

In this paper, we have studied this method in more depth and have applied it to medical records in order to test its effectiveness and usefulness in such a heterogeneous field.

#### 1.1.2. Tag Cloud in Databases

In [23], a graphical interface for browsing and querying in a primary care database has been developed. The main advantage of this system is that all kinds of users would be able to handle it, including non-experts.

The tag cloud can have the same function, but it wins in terms of simplicity. The tags can be displayed within the tag cloud with different font sizes depending on their frequency of use [7]. These different font sizes provide semantic expression to the tag cloud and highlight the most relevant terms [24].

Tagging is widely used in text mining in biomedical texts [18] in order to simultaneously obtain grammar and meaning. However, in databases, this type of collaborative tagging is not easily applicable. Some authors [25,26] have previously used tag clouds in databases to summarize query results. However, our main objective is to represent text content, facilitating the visual exploration of the information, although our tag cloud can also work as a query assistant.

Other authors [27,28] have represented the global content of the database by using a tag cloud in which the tags are composed of single terms, whilst our tag cloud allows the use of multi-terms in tags.

## 2. Materials and Methods

Figure 1 highlights the different stages of the entire experimentation process.

Next, the materials and methods applied in the process are explained.

### 2.1. Dataset

For experimental evaluation, we have used anonymous data from the electronic records of the Hospital Clínico San Cecilio de Granada in Spain. The data are in Spanish, and the data are stored in a set of tables in a relational database. We have selected, as a starting point, the table that gathers together the information referring to the surgical interventions and that includes 24,481 records. The main attributes of this table are “Diagnostics” and “Proposed Intervention”, where the content is a short text comprising one or more sentences.

### 2.2. Methodology for the Tag Cloud Generation

In order to confront the difficulties found in medical databases with unstructured texts described in Section 1, we propose the tag cloud as a visual interface. The tag cloud represents and summarizes the text; allows concept identification, their relevance and relationships; and facilitates browsing and querying.

The mathematical model proposed to define a multi-term tag cloud is based on the concept of the WAPO-Structure and its properties (see Section 2.2.2). The WAPO-Structure preserves the semantics of the text by allowing related terms to remain ordered and to remain together, a very important feature in the visualization of the tag cloud. It is generated from a method based on the co-occurrence of sequences in the text [22]. Once generated, the structure represents the content of the information, serving as a tool for identifying concepts. Its visualization through the tag cloud allows non-expert users or those without prior knowledge of the text to make queries intuitively.

Next, we explain the complete methodology followed to obtain a tag cloud from a textual attribute in a database. Figure 2 summarizes the different phases of the process, which are developed in the following steps: preprocessing, generation of the intermediate form, post-processing, and visualization.

#### 2.2.1. Preprocessing

First, syntactic preprocessing is carried out by applying tokenization filters, removing the stop words, and performing simple stemming to eliminate gender and plural forms from the raw data.

Subsequently, semantic preprocessing is applied in which auxiliary files are created with the help of experts and contain the synonyms and acronyms of specific hospital language.

For example, the term “IZQUIERDO” for which its English translation is “LEFT” appears in the original text with all the following forms (considered synonymous in the auxiliary file): IZQ, IZDA, IZDO, IZQD, IZQDO, IZQDA, IZ, IZQIERDO, IZQU, IZQUI, IZQUERDO, HIZQ, IZQ1-UIERDA, IZQUIE, IZQUIEDO, IZQUIER, IZQUIERA, IZQUIERD, IZQUIERDOP, IZQUIERTDO, IZQUIRDO, IZQUUIERDO, and IZQUIERDA. This is without considering the different variants when mixed uppercase and lowercase occurs.

The acronyms file used fulfills the main function of obtaining all the acronyms referring to the same concept with the same syntactic form. Table 1 shows a sample of the acronyms in such file.

Semantic preprocessing basically consists of removing the separation points between the letters in the acronyms and replacing the synonyms found in the original text by a canonical representative form (in the example observed for “IZQUIERDO”, all the different forms of the term were replaced by a single one: “IZQUIERDO”).

Table 2 shows an example of clean text after applying syntactic and semantic preprocessing.

#### 2.2.2. Generation of the Intermediate Form of Representation

In order to extract WAPO-Structure [29], we first calculated the support of the terms as the proportion of transactions containing that term, and then the text is searched by looking for frequent terms according to a given minimum support. Once located, they combine with each other to give rise to the candidate sequences of two terms, and those that are frequent in the text remerge with each other, maintaining the strict order of adjacency between terms, to give rise to the candidate sequences of three terms and so on until the frequent sequences with the greatest number of terms are found, which are called the spanning sequences.

##### WAPO-Structure

Let $X=\{x_{1},x_{2},\ldots ,x_{n}\}$ be a referential set of items and S={A,B,…} a set of frequent weighted sequences with a cardinal higher than or equal to one, and A,B,… are weighted AP-Sequences such as the following.
(1)∀A,B∈S;A⊈B,B⊈AandB≠A.

A WAPO-Structure generated by S,E=g(A,B,…) is the set of AP-Sequences in which its spanning sequences are A,B, and so on.

*Note.* If *A* is a spanning sequence of the WAPO-Structure, then all of its sub-sequences will be frequent and belong to the WAPO-Structure.

Each AP-Sequence in the WAPO-Structure will be the generated from a spanning sequence as the following example shows.

##### Example of an AP-Sequence

Let us suppose that the sequence {amputatión dedo pie izquierdo} is a spanning sequence, then a AP-Sequence is generated from it as shown in Figure 3.

The number that appears next to each sequence of terms indicates the absolute frequency (weight) of this sequence in the text.

The WAPO-Structure is the set of all the AP-Sequences generated from all the spanning sequences found in the text.

In order to decide the support value for our experiment, a group of experts belonging to our research group helped us to carry out a trial and error analysis. Small values for the support make the number of terms in the visualization too large, which prevents them from being clearly distinguished. For large values, important terms are lost in the visualization, so the most adequate support is that which keeps an acceptable number of terms in the visualization but not so many as to make identification difficult.

Finally, we have selected the structure corresponding with a support equal to 0.3%.

WAPO-Structures provide some methods to manipulate them and their spanning sequences. The operations provided are described in detail in [29]. In addition, an alternative generation method is presented in [30].

#### 2.2.3. Post-Processing

Once all the information from the frequent itemsets is obtained, including their weight and the WAPO-Structure generated, we perform post-processing. First, we use part of speech tagging. Nouns are good candidates to be represented in the tag cloud, but verbs and adverbs are not especially useful words for representing the content of information in this specific field. For this reason, in the post-processing stage, we remove all itemsets that contain verbs or adverbs. Adjectives by themselves are not informative, but if they are next to a noun, then they modify it by adding semantics.

By considering these observations, a set of rules has been defined to determine when an itemset meets the requirements for its visualization within the tag cloud according to its grammatical category. The rules are the following for the Spanish language:Level one itemsets: The good candidates are nouns (N);Level two itemsets: The good candidates are those composed of two nouns (NN) or of an adjective and a noun: (AN) and (NA);Level n itemsets: The good candidates are those composed of a valid combination of terms in the previous levels plus a noun or an adjective. For level three, it would be the following: (NNN), (ANN), (NNA), (NAN), (ANN), (AAN), and (ANA).

In addition to the frequent itemsets that contain verbs or adverbs, all those that do not meet any of these rules are removed. The rest are displayed in the corresponding tag cloud.

#### 2.2.4. Visualization

After post-processing, colour is applied to the tags in order to make it easier to distinguish them. We have chosen black for mono-terms and red shades for the multi-terms. In this manner, an aesthetic tag cloud is achieved in which the colours have functionality.

In Figure 4, we can observe the visualization of the example in Figure 3.

### 2.3. Retrieval System Implementation

A retrieval system has been implemented where the tags work as queries. The textual records of the attribute have first been tagged with the most appropriate tags of the cloud in order to know which ones should be recovered with each of these tags.

As time constraints do not allow us to manually tag the 24,481 records, we set a relative error of 5% for a 95% of confidence in the mean estimation, which provides us with a sample size close to 500 records that have been randomly selected.

The tagging has been carried out by a group of experts in the medical field. They have been provided with the records of the sample and a tag cloud generated from these records which contains the tags to use. It is important to highlight that the entire tag cloud generation process is automatic, and this manual tagging is only performed to evaluate the tag cloud according to the opinion of the experts.

### 2.4. Metrics for Evaluation

First, precision (fraction of correct results among all the results retrieved) and recall (fraction of correct results among all the results that should have been retrieved) have been calculated, as well as the *F*1-score that is the harmonic mean between them.

Next, we calculate coverage, overlap, and balance metrics with the formulas provided in [31]. The coverage provides us with the fraction of the original text represented by the terms in the tag cloud. This metric takes values in the range [0,1]. A value close to one indicates that the terms in the tag cloud represent most of the original text. The overlap determines the degree to which different terms in the tag cloud represent the same information in the original text. This metric also takes values in the range [0,1]. A value close to zero indicates that the degree of overlap is low, so the terms represent different information. The balance is a metric related to the number of results retrieved by the tags. It is said that a tag cloud is balanced if its tags retrieve a similar number of results. This metric takes values in the range [0,1] as well. A value close to one indicates that the tag cloud is balanced. The average coverage per tag has also been calculated, and it is understood as the fraction of the original text represented, on average, by each tag in the tag cloud.

### 2.5. Satisfaction Survey

In order to evaluate the degree of user satisfaction with the tag cloud, we carried out a survey.

#### 2.5.1. Procedure

We contacted several specialists from different areas (cardiology, obstetrics and gynecology, and dermatology, etc.) and requested for collaboration by email using the snowball sampling. This type of sampling is widely used when there is no base, which frequently occurs in the medical field [32,33]. It is defined as a non-probability sampling technique in which existing subjects provide referrals to recruit others participants, where it is necessary to have one or more subjects as a starting point that provide other contacts. These first subjects are called the “seeds”. This sampling technique can go on and on, which is similar to a snowball increasing in size until the point where a researcher has enough data to analyze.

In total, we contacted 60 specialists, out of which 23 were willing to collaborate. They all came from the same hospital that collaborated in the data transfer, which is Hospital Clínico San Cecilio de Granada in Spain, and they were provided with a link to a website where we had enabled the tool (the Retrieval System is explained in Section 2.3) for experimentation and subsequent evaluation. They had to test the usability, information retrieval ability, ease of concept identification, search utility, and content representation capacity and answered the questions of the survey in what would be the evaluation process. For this purpose, we embedded a form on this same website with some brief instructions and some statements with which the participants had to express their degree of agreement.

The first four statements were the following:1.The tag cloud presented seems intuitive and easy to use.2.The tag cloud presented provides information about the content of the database.3.The information retrieved with the tags is consistent with them.4.A tag cloud similar to the one presented would help me to search a medical database.

The last four are related to the ease in identifying concepts. They are formulated in the following manner: “(5–8). It is easy for me to identify a concept in the tag cloud related to ...(*definitions in Table 3*)”.

We tried to evaluate that the user was able to identify concepts located in different locations of the visualization and with different font sizes, therefore, that was the reason for specifically choosing those four concepts. For example, the term “Cesarea” has a big size, and it is placed near the top of the visualization, while the term “Artroplastia total” has a lower size, and it is located in the low middle.

The first statement is related to the simplicity of the tag cloud, the second one is related to content representation, the third and fourth are related to the retrieval system implemented over the tag cloud, and the last four are for evaluating the ease of the identification of terms.

The degree of agreement with these statements is expressed by using the Likert scale with a numerical rating from 1 to 5, where 1 indicates “complete disagreement” and 5 represents “complete agreement”.

Based on this classification, the correspondences of the degree of agreement with the given rate are as follows:1: Complete disagreement;2: Disagree;3: Indifferent;4: Agree;5: Complete agreement.

Finally, the participants were asked to provide some suggestions about how the different aspects can be improved.

#### 2.5.2. Software

Statistical analysis has been performed with StatGraphics Centurion XV [34], IBM SPSS Statistic 23 [35], and G*Power [36].

#### 2.5.3. Participants

The minimum sample size required to obtain an absolute error below 0.5 points in mean estimation assuming normality is **18 participants**, considering a standard deviation equal to 1 (Range/4 [37]) and a 95% of confidence. This result has been obtained by using the statistical software StatGraphics Centurion XV [34], which makes use of the following formula:(2)$$n>{\left(\frac{t_{n-1;\frac{\alpha }{2}}\cdot S_{d}}{e}\right)}^{2}$$
where $S_{d}=1$ is the estimated sample deviation, e=0.5 is the error, α=0.05 is the level of significance, and $t_{n-1;\frac{\alpha }{2}}$ is the quantile $\frac{\alpha }{2}$ of the Student’s t-distribution with n−1 degree of freedom, which corresponds to a 95% confidence interval. Since this quantile is dependent on *n*, it can be approximated to the normal quantile for large sample sizes (more than 30 in practice) [38]:(3)$$n>{\left(\frac{z_{\frac{\alpha }{2}}\cdot S_{d}}{e}\right)}^{2}$$
where $z_{\frac{\alpha }{2}}=1.96$, providing a minimum sample size of 16 participants. This approximation is not appropriate for small samples, and StatGraphics Centurion XV does not need to perform it it, obtaining a more precise result with the quantile of the Student’s t-distribution, which produces a sample size of 18 participants.

In total, we have information from 23 anonymous participants with training and experience in different areas of medicine, which has reduced the minimum error previously set.

In hypothesis testing, this final sample size corresponds with a value of power equal to 0.956 for a large effect size (0.8) and equal to 0.630 for a medium effect size (0.5) where the critical t-value is equal to 2.074 (calculation obtained with G*Power). However, the power and effect sizes are more appropriate in comparing the mean between two groups than in estimating the mean value for one sample [37,39].

## 3. Results

We generated a tag cloud from the attribute “Proposed Intervention” (see Figure 5). This attribute is especially complex since it contains a large number of syntactic variations, and the information in it was introduced by different practitioners in natural language.

This tag cloud works as a retrieval system, and we have used it as the tool for experimentation and subsequent evaluation.

### 3.1. Metrics Calculation

Table 4 shows the average of the precision, recall, and F1-score by considering all the tags in the generated cloud (see Figure 5). Two types of query have been considered for the calculation:Type I. This retrieves all the entries where the terms appear in the same strict adjacency order as they did on the tags;Type II. This retrieves all the entries where the terms appear in the same order as had in the tags, without considering strict adjacency.

The values obtained for precision, recall, and *F*1-score are very good in both types of query, with type II being a little higher.

For precision, a higher value than for recall is achieved. This means that almost all the entries that are recovered are relevant, although there is a small proportion of relevant entries that were not recovered. We can solve this issue by increasing the support of the tag cloud, but an increase in the recall would probably decrease the precision. Furthermore, for recovering the records that remain inaccessible from the tag cloud, the system can also be queried with the traditional method.

Table 5 shows the values obtained for coverage, coverage per tag, overlap, and balance.

Being such an extensive and heterogeneous dataset, a coverage close to 60% is a pretty good value. It is possible to increase this value by decreasing the support and allowing a higher number of tags in the visualization. In other experiments, we obtained better coverage with our automatic process than other approaches where the tag cloud was manually built over the same database [29]. The overlap is practically zero; thus, different tags represent different information. Finally, the value obtained in the metric of balance indicates that the tag cloud is unbalanced, which provides this type of visualization with the ability to highlight the most relevant topics.

### 3.2. Statistical Analysis of the Survey Results

The hypothesis that the assessments about the ease of concept identification is independent with respect to the concepts provided that it is verified by using a chi-square independence test [40]. For this test, we have used the Likelihood Ratio as the discrepancy measure since a 60% of the cells have an expected frequency of less than five, and there are some cells with an expected frequency equal to zero. The experimental value for the Likelihood Ratio is 9.16 with a *p*-value equal to 0.6888, so we can assume that the rates obtained in the concept identification are independent to the concepts provided with a significance of 5%; therefore, the concepts of identification rates are merged into only one variable.

By taking into account the rest of the assessments, there are five variables to analyze in total:1.Ease of Use: Evaluates if the tag cloud is intuitive and easy to use;2.Identification: Evaluates whether the identification of concepts is simple in a global manner;3.Retrieval: Determines if the information retrieved is consistent with the tags;4.Representation: Checks if the tag cloud represents the content properly;5.Search Utility: Evaluates if users would use the tool to conduct searches in medical databases.

Figure 6 shows the bar graphs and the pie charts for each of these variables. When looking at the graphs, it is easy to realize that the highest percentages correspond to rates four and five (agree and complete agreement) for the first four variables. For the variable *Search utility*, the highest percentage is for rate three (indifferent), but this does not exceed the sum of the percentages corresponding to rates four and five. In all cases, more than 60% of respondents provided positive ratings (four or five), and this percentage is much higher for the first three variables, reaching almost 90% for *Ease of use* and surpassing it with *Retrieval*.

In the mean graph (Figure 7) the confidence intervals for the mean of the rates can be observed, indicating the mean or average with a small red cross and the confidence intervals with a blue line. The interval corresponding to *Ease of use* is the one with highest values. All the intervals have values greater than three, which means that the average of the rates is greater than three in all cases for a 95% confidence; thus, the participants would agree with all the assessments made.

To verify this result, a unilateral test for the mean of a sample is performed for each of the five variables. In the null hypothesis, the mean is less than or equal to three, and it is higher in the alternative. In the case of rejecting the null hypothesis in favour of the alternative, we demonstrate that, on average, the participants agree with the statements made about the capacity of the tag cloud. To carry out this test, normality and equality of variances have been assumed. Table 6 shows the summary of results.

Therefore, it can be said that the mean of all the variables is higher than three; thus, the participants agree with the statements made about the capacity of the tag cloud with a significance of 5%:1.Is intuitive and easy to use;2.Represent the content appropriately;3.Retrieve consistent information;4.Help in carrying out searches;5.Concept identification is easy.

## 4. Discussion

We have applied a simple method based on co-occurrences in order to extract information from unstructured texts in a medical database and obtained very high precision, which does not happen with other methods based on co-occurrences. This is due to taking into account not only the isolated terms but also the sequences.

The F1-score obtained is similar to the best learning-based methods [17,41], both for exact and inexact matching, with our dataset being much bigger. We obtained more precision than these methods and less recall, but the recall values could be increased simply by adding a larger number of terms to the tag cloud and decreasing the support. We have also improved the precision obtained in [21] by taking into account the order in the terms. The fact of not manually tagging the text in the tag cloud generation decreases the granularity found in other systems, such as the one observed in [42].

The coverage metric, which is close to 60%, indicates that the tag cloud represents a good fraction of the original text, with the overlap being close to zero, which indicates that the tags represent different information. The balance is also close to zero and is a good value for a tag cloud where the main objective is to be able to highlight the most relevant concepts; thus, the amount of results that retrieved with each tag would be unbalanced. The coverage could be increased by increasing the number of terms in the tag cloud, but this would reduce their visibility, and the value obtained is actually quite good considering the heterogeneity of the text. In addition, to recover the records that remain inaccessible from the tag cloud, the system could be queried by using the traditional method.

The satisfaction survey carried out with expert users verifies that, on average, they think that the tag cloud is an intuitive and user-friendly tool that provides information about the content of the database, which retrieves information consistent with the tags that compose it, which helps to search in a medical database and facilitates the identification concept. The great advantage of this tag cloud is that it does not require human intervention for its generation.

### Limitations

We have only tested our approach on short texts, and it has yet to be proved that this approach also works on larger documents. Therefore, when the recall is increased, the support has to be increased, generating large tag clouds that may not be displayed correctly on small screens.

## 5. Conclusions and Future Work

As we already know, it is difficult to process and access the information stored in large clinical databases. Graphical interfaces that help to query the database and help in representing its content produce numerous deficiencies and are often complicated to use when providing query results, which usually have low precision.

The main contributions of this study include the definition of a method for information processing that would work well in a large and heterogeneous clinical database, preserving the semantics of the textual attributes, and a graphical interface that is easy to use and that properly represents content information. This graphical interface helps to query the database and obtains query results with high precision. We have provided a complete evaluation of the interface by calculating the appropriate metrics and by using a survey of expert users with satisfactory results.

Some of the limitations of our proposal are in relation with the screen resolution, the human visual system resolution, and the limits of available computational resources. To confront these limitations, we can apply some of the strategies proposed in [3] as future work. Furthermore, we will consider the generation of tag clouds in additional databases as well as the creation of a multi-language system for the generation of tag clouds based on ontologies.

Another idea is to have a multi-level tag cloud using an ontology where choosing one item allows viewing others in the entire category or narrowing down the query. This would reduce the size of the cloud and allow for a combination with processes such as a faceted search. It would also be interesting to offer a multilingual tool so that non-Spanish-speaking researchers can also access the database. We are also contemplating the application of clustering techniques in our data as well as focusing on the search of entities by using second-level text mining.

## Figures and Tables

**Figure 1 entropy-23-01275-f001:**
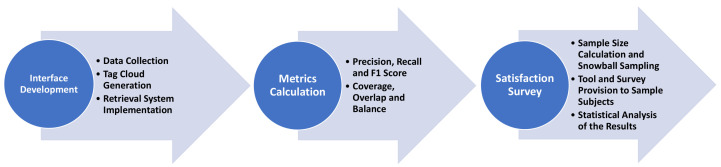
Stages of the experimentation process.

**Figure 2 entropy-23-01275-f002:**
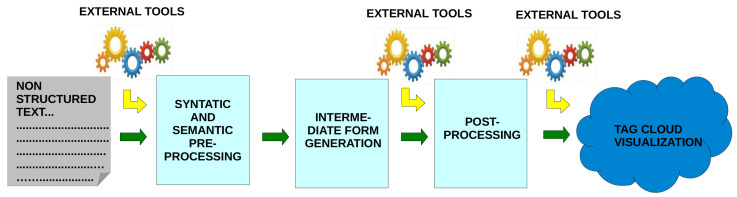
Phases of the methodological process.

**Figure 3 entropy-23-01275-f003:**
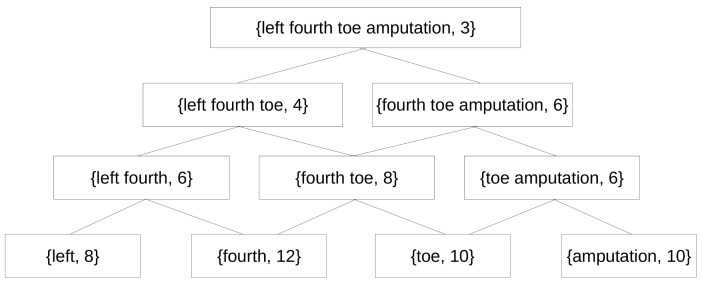
AP-Sequence generated from the spanning sequence *{left fourth toe amputation}*.

**Figure 4 entropy-23-01275-f004:**
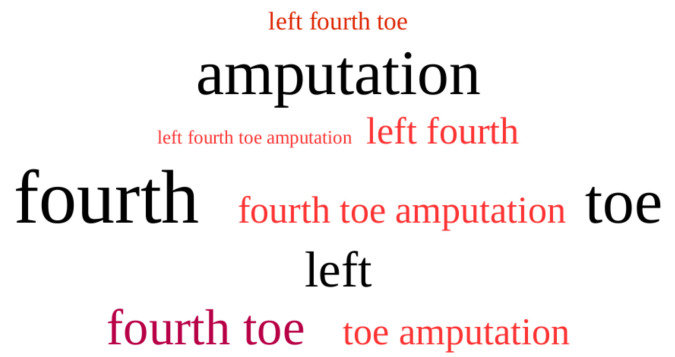
Visualization of the AP-Sequence in Figure 3.

**Figure 5 entropy-23-01275-f005:**
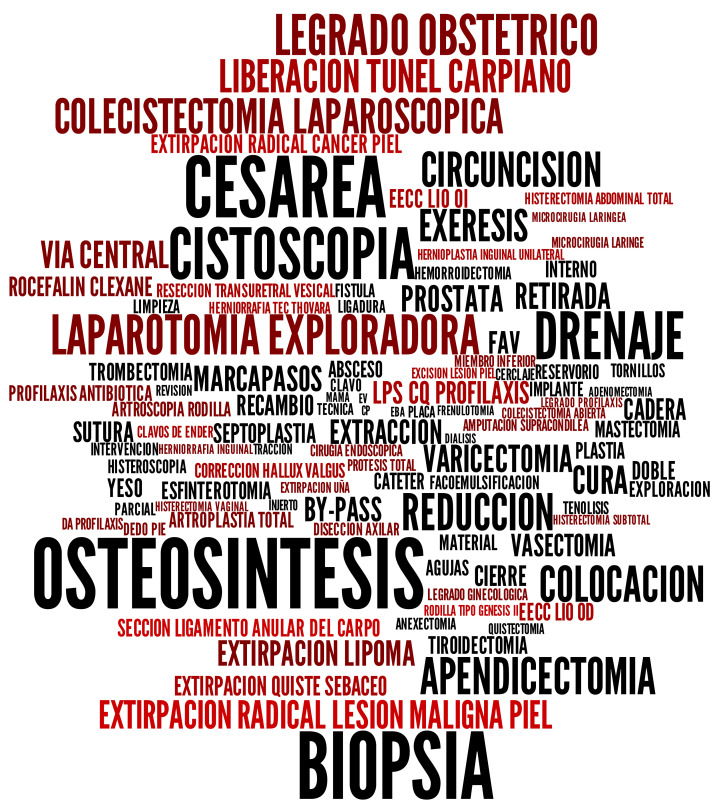
Tag cloud after postprocessing and colour with 0.3% of support.

**Figure 6 entropy-23-01275-f006:**
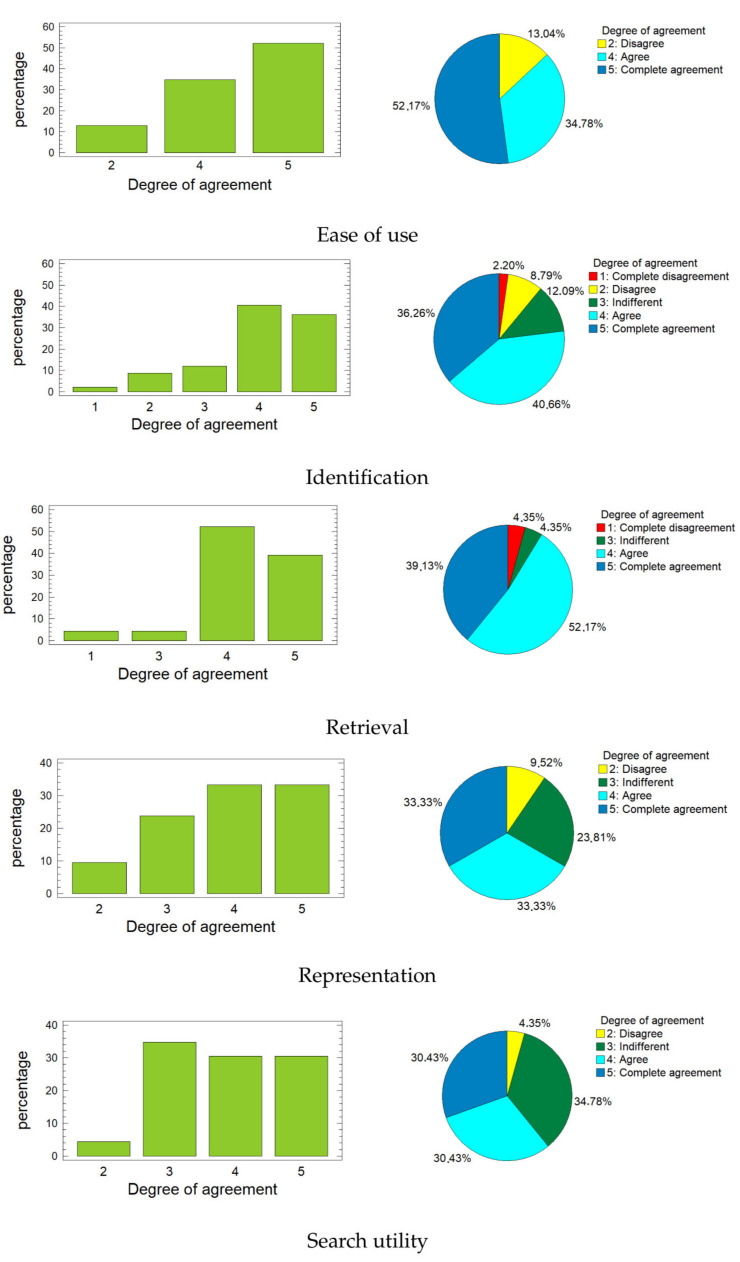
Bar graphs and the pie charts for each of the variables.

**Figure 7 entropy-23-01275-f007:**
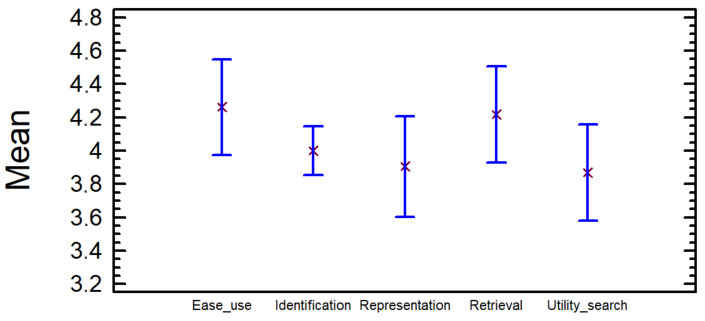
Mean graph with confidence intervals at 95% for the degree of agreement (from one to five) with different assertions about the characteristics of the tag cloud.

**Table 1 entropy-23-01275-t001:** Examples of acronyms found in the original text.

Acronyms	Terms	English Traslation
EECC	Extracción extracapsular del cristalino	Extracapsular cristalline lens extraction
LIO	Lente intraocular	Intraocular lens
OI	Ojo izquierdo	Left eye
EBA	Exploración bajo anestesia	Examination under anesthesia
CP	Cámara posterior	Posterior chamber
PA	Peritonitis aguda	Acute peritonitis

**Table 2 entropy-23-01275-t002:** Example of the text after cleaning.

Short Text	Modified Short Text	English Traslation
AMPUTACION 4 DEDO PIE IZDO	AMPUTACION 4 DEDO PIE IZQUIERDO	Left fourth toe amputation
ENDORREDUCCION DE F.A.V.	ENDORREDUCCION DE FAV	Reduction in arteriovenous fistula
legrado	LEGRADO	Cuterage
Mastectomia mas D.A.	MASTECTOMIA MAS DA	Mastectomy plus axillary dissection

**Table 3 entropy-23-01275-t003:** Concept definition and expected objects to be identified in a tag cloud in Figure 5.

	Definitions Provided in Statements 5–8	Term
1	Surgical intervention for the birth of a baby	Cesárea
2	Surgical abortion or treatment after abortion	Legrado obstétrico
3	Surgical operation aimed at complete	Artroplastia total
	reconstruction of an obstructed or ankylosed joint	
4	Technique widely used in the operation of cataracts	Facoemulsificación

**Table 4 entropy-23-01275-t004:** Average Precision, Recall, and $F_{1}$-score for the tag cloud in Figure 5.

Query Types	Precision	Recall	$F_{1}$-Score
Type I	0.940	0.780	0.823
Type II	0.990	0.870	0.904

**Table 5 entropy-23-01275-t005:** Coverage, overlap, and balance of the tag cloud in Figure 5.

Coverage	Coverage Per Tag	Overlap	Balance
0.58793	0.00706	0.00003	0.05022

**Table 6 entropy-23-01275-t006:** Summary result of the hypothesis tests.

Verification That the Mean of the Degree of Agreement with the Assessments
About the Capabilities of the Tag Cloud Can Be Considered Higher Than 3,
That Is, the Participants Agree with These Assessments.
Hypothesis test for the mean of a population:
$H_{0}:\mu \leq 3$, $H_{1}:\mu >3$, α=5%
•*The tag cloud presented is intuitive and easy to use.*
$t_{exp}=5.98804$, *p*-value=$2.5\cdot 10^{-6}$⇒ Reject $H_{0}\;$ ⇒
Agree with the assessment. Sample mean = 4.26087
•*It is easy for me to identify the concepts provided in the tag cloud.*
$t_{exp}=9.33422$, *p*-value=0.0 ⇒ Reject $H_{0}\;$ ⇒
Agree with the assessment. Sample mean = 3.96739
•*The information retrieved with the tags is consistent with them.*
$t_{exp}=6.47025$, *p*-value=$8.25\cdot 10^{-7}$⇒ Reject $H_{0}\;$ ⇒
Agree with the assessment. Sample mean = 4.21739
•*The tag cloud presented provides information about the content of the database.*
$t_{exp}=4.16603$, *p*-value=$2.3\cdot 10^{-4}$⇒ Reject $H_{0}\;$ ⇒
Agree with the assessment. Sample mean = 3.90476
•*A tag cloud similar to the one presented would help me to search a medical database.*
$t_{exp}=4.5344$, *p*-value=$8.1\cdot 10^{-5}$⇒ Reject $H_{0}\;$ ⇒
Agree with the assessment. Sample mean = 3.86957

## Data Availability

We do not have permission to share the data.

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
