# Peer review of "On Building and Evaluating a Medical Records Exploration Interface Using Text Mining Techniques"

_entropy, 2021, doi:10.3390/e23101275_

Round 1
Reviewer 1 Report
The aim of this study was the visual exploration of the information in medical databases using multi-term tag clouds as content representation tools and as assistants for the browsing and querying tasks. Whilst this manuscript has some merit, the structure is very confusing such as methods mixed with background and the manuscript requires extensive re-structuring to make it easier for the reader to follow and comprehend. The title and abstract do not fully represent the intent of the study as this study also included survey data of satisfaction. As an English reader, it was difficult to understand some of the tables and figures and further clarification may be worthwhile to appeal to a broad readership. The survey analysis is lacking scientific and statistical merit and requires further clarification. This component appears to be more like a qualitative than quantitative exercise.
Please correct your English throughout the manuscript (e.g. LINE 17 “which is complex task”)
INTRODUCTION: This section lacks a clear structure and was rather confusing to the reader. Thus, it is highly recommended it is restructured to discuss the main focus of this paper where the aim is clearly articulated. This section includes information that would normally be contained in the methods and discussion.
LINES 21-30: Please clarify the main message from this paragraph as it was not clear to the reader. Is this paragraph about identifying errors or providing structure or a simple interface?
LINES 32-34: Your statement that “tag cloud as graphical interface that…. allowing the identification of entities and their relationships” is not technically correct as the cloud does not provide the links to relationship (e.g. like in network analysis).
LINES 39-40: Please clarify your statement “and in other studies with a similar purpose to ours but with a totally different generation method, resulting in a mono-term tag cloud” as it is unclear to the reader.
LINES 49-52: This paragraph reads as though it is a limitation of your study. If so, then this should be included in your Discussion rather than your Introduction.
MATERIALS AND METHODS:
LINES 104-117: This relates to the methods so it was unclear why this is outside of Section 2.1. It would be much clearer to the reader if you please provide a flowchart of your methods and then provide the detail.
Tables 1 & 2, Figure 1: This is not in English so is very confusing to the non-English reader.
2.2. Please outline your data prior to your methods (i.e. materials, then methods). Was Ethics approval required to access this data. If so, please outline the approval.
2.4.1. Where did the ” several specialists requesting for collaboration by email”? What was the sampling methodology? What was the base? What was the profile of this cohort? This seems more like a qualitative exercise rather than a survey.
LINES 238-239: Please provide more detail about your “tool for experimentation and subsequent evaluation” What was the ‘experiment’ component? Please provide more detail about the evaluation process.
METHODS: Please ensure you provide the software used to perform your analysis (and references).
If you are using Precision, Recall and F1- score, then also please provide Accuracy.
TABLES: Please ensure you add your tables directly after they are mentioned in the text of your manuscript (e.g. LINE 276 “Table 4”)
FIGURE 4: Please label your x and y axis in your charts.
LINES 304-305: What does “Chi-square independence test, with a discrepancy measure equal to 9.16 and a p-value equal to 0.6888” Why have you set your p-value to 0.6888?
LINE 324: Please clarify what the ‘mean’ measure is in your “In the mean graph…” statement and clarify it on the associated graph.
Author Response
The authors would like to thank the Reviewer 1 for their valuable comments and suggestions that have greatly helped us to improve the paper.
Please see the attachment with the responses to the comments.

Reviewer 2 Report
This work presents a set of methods for processing information from large clinical databases, following a number of defined sets of semantic of textual (i.e. syntactic) attributes. The authors then present a graphical interface to represents the main content information for visualization purposes with high precision (their proposal is to use tag cloud). The authors provided an evaluation of the interface through the calculation of various metrics and a performed survey of expert users, the presented results were characterized as satisfactory.
This work is interested but it does not clearly state its purpose and originality as well as innovation in terms of scientific breakthrough. It is not clear what is the problem and how is this solved (i.e. the database already use semantic annotation – the innovation is clearly more on the representation of the created “meta” knowledge of the collected data). In terms of UI then we agree that the results are of course positive but a standard method (ie. SUS or any other would be nice to be used to show significance of the study)
Author Response
The authors would like to thank the Reviewer 2 for their valuable comments and suggestions that have greatly helped us to improve the paper.
Please see the attachment with the responses to the comments.

Reviewer 3 Report
Overall, I feel this is a very interesting article. It has not only looked into developing a visual interface to make sense of vast medical records but also considered domain experts' perception concerning that and compared it with the tool's outcome. Thus, technically this is solid work. Having said that, some improvements in the presentation are needed.
First, not until late in the article, is apparent that the article focuses on the Spanish dataset. Consequently, a non-Spanish reader may struggle to understand some earlier parts like the pre-processing in 2.1.1. Perhaps some explanations for English-based readers concerning the language issues will be good.
Also, some reorganisations can be there to make the paper more readable. Not all readers will have knowledge of the covered domain. Like, the paper speaks of WAPRO structure in the early part of section 2, but its details are provided in 2.1.2. Thus, to avoid confusion to the readers, the authors may consider bringing the section forward or at least suggest to the readers that detail of the structure are outlined in a later part of the paper.
As early as section 1, the paper speaks of technical terms like recall and accuracy, and then in section 2.3 speak of coverage, overlap, and balance. But what do these terms mean in this specific research context? - this is a question that may come up in the mind of a reader not well familiar with the domain. Further, some of the concepts that the article use like support and frequent itemset are also considered in the machine learning domain, especially association rule mining. Machine learning also has a specific definition of F1-score. Consequently, unless a reader is familiar, he/she may feel a bit of struggle to follow the article. Thus, the authors may consider clarifying these terms before the first use.
On the same note, the authors may consider putting a figure to highlight the steps/stages in the experimentation - like, developing the interface, then doing a survey.
Also, in line 251: the phrase "their different locations in the visualization and their different sizes" - is unclear. May explain this a bit more.
Lastly, a minor grammar issue: line 32: may change "offer" to "offering"
Author Response
The authors would like to thank the Reviewer 3 for their valuable comments and
suggestions that have greatly helped us to improve the paper.
Please see the attachment with the responses to the comments.

Round 2
Reviewer 1 Report
Thank you for the modifications to your manuscript. However, there are still a number of concerns requiring attention.
ABSTRACT: correct 'question' to 'questions'
MANUSCRIPT: Suggest further checking of the English (e.g. lines 272 and 289 should be in the past tense, line 325 should have a full stop at the end of the sentence).
POWER ANALYSIS: Lines 312-314: This power analysis for the correct sample size does not seem complete nor correct. If you calculate the estimated sample size for a one-sample mean test with the study parameters: alpha = 0.05, power = 0.80, delta = 0.5, mean difference of diff = 0.5, sd = 1.0, the estimated sample size: N = 34. With power set to 0.90 N = 44, and power set to 0.95 N = 54. Please clarify your calculations for your survey sample size?
Section 2.5.1. Procedure: Please address my original request and explain what the “specialists” specialised in? How many were invited? How many agreed?
Figure 6: Please relabel the Rates to “Degree of agreement” as per your methods and also label rates 1 to “Complete disagreement” and 5 to "Complete agreement” so the messages in these figures are clearer to the reader.
Unfortunately, my original question regarding the chi-square test was mis-interpreted as it read as methods and not results. Since you ran aa chi-square test then there are strict assumptions for this test -> please see “The Chi-square test of independence” by Mary L. McHugh, Biochem Med (Zagreb). 2013 Jun; 23(2): 143–149. doi: 10.11613/BM.2013.018. Please ensure you have documented this test in your METHODS and that no assumptions are violated, especially “The value of the cell expecteds should be 5 or more in at least 80% of the cells, and no cell should have an expected of less than one”
Figure 7: Please ensure the font size is consistent and ensure the ‘red cross’ is identifiable.
Author Response
The authors would like to thank the Reviewer 1 for their valuable comments and suggestions that have greatly helped us to improve the paper.
Please see the attachment.

Reviewer 2 Report
No comment
Author Response
The article has been modified, adding more complete information in some parts and correcting some grammatical errors. We hope it has improved now.
Round 3
Reviewer 1 Report
Thank you for addressing my comments.